# Associations between *β*-Lactamase Types of *Acinetobacter baumannii* and Antimicrobial Resistance

**DOI:** 10.3390/medicina59081386

**Published:** 2023-07-28

**Authors:** Kristina Černiauskienė, Asta Dambrauskienė, Astra Vitkauskienė

**Affiliations:** Department of Laboratory Medicine, Faculty of Medicine, Medical Academy, Lithuanian University of Health Science, Eivenių g. 2, LT-50161 Kaunas, Lithuania; kristina.cerniauskiene@lsmu.lt (K.Č.); asta.dambrauskiene@lsmuni.lt (A.D.)

**Keywords:** *Acinetobacter baumannii*, extended spectrum *β*-lactamase, AmpC *β*-lactamase, multidrug resistance

## Abstract

*Background and objective*: *Acinetobacter baumannii* (*A. baumannii*) is an important nosocomial pathogen that not only possesses intrinsic resistance to many classes of antibiotics, but is also capable of rapidly developing antimicrobial resistance during treatment. The aim of this study was to determine the characteristics of resistance of *A. baumannii* strains to *β*-lactams and other tested antibiotics, to evaluate the associations between the phenotypes of resistance to *β*-lactams and other tested antibiotics, and to evaluate the changes in antibiotic resistance of *A. baumannii* strains over 5 years by comparing the periods of 2016–2017 and 2020–2021. *Materials and methods:* A total of 233 *A. baumannii* strains were isolated from different clinical specimens of patients treated at the Hospital of Lithuanian University of Health Sciences in 2016–2017 (*n* = 130) and 2021–2022 (*n* = 103). All clinical cultures positive for *A. baumannii* were analyzed. The type of *β*-lactamase was detected by phenotypic methods using ESBL plus AmpC screen disk tests and the combination meropenem disk test. *Results:* In both periods, all *A. baumannii* strains were resistant to ciprofloxacin; resistance to carbapenems, piperacillin/tazobactam, gentamicin, and tobramycin was noted in more than 80% of strains. A comparison of two periods showed that the percentages of *A. baumannii* strains producing two or three types of *β*-lactamases were significantly greater in 2021–2022 than in 2016–2017 (94.2% and 5.8% vs. 17.7% and 2.3%, respectively, *p* < 0.001). Isolates producing two or three types of *β*-lactamases were more often resistant to tigecycline, tetracycline, and doxycycline than strains producing one type of *β*-lactamase (*p* < 0.001). *Conclusions:* The frequency of isolation of *A. baumannii* strains producing two different types of *β*-lactamases (AmpC plus KPC, AmpC plus ESBL, or ESBL plus KPC) or three types of *β*-lactamases (AmpC, KPC, and ESBL) and the resistance rates to ampicillin/sulbactam, tigecycline, tetracycline, and doxycycline were significantly greater in 2020–2021 as compared with 2016–2017. The production of two or three types of *β*-lactamases by *A. baumannii* strains was associated with higher resistance rates to tetracyclines.

## 1. Introduction

*Acinetobacter baumannii* (*A. baumannii*) is an obligate aerobic, nonfermenting, Gram-negative nonmotile bacterium that was discovered by Dutch microbiologist Martinus Willem Beigerinck in 1911 [1]. The ability of the microorganism to survive in various environmental conditions and to last longer on wet or dry surfaces has turned it into an endemic, health care–associated pathogen and a common cause of infection outbreaks [2,3]. People who have a weakened immune system, chronic lung disease, or diabetes are at a greater risk of being more susceptible to *A. baumannii*–caused infections, especially patients with a prolonged hospital stay, those with open wounds, or anyone with urinary or other catheters [4].

*A. baumannii* is an important nosocomial pathogen that not only possesses intrinsic resistance to many classes of antibiotics, but is also capable of rapidly developing antimicrobial resistance during treatment. This pathogen has become resistant to almost all antimicrobial agents currently available, including aminoglycosides, quinolones, and broad-spectrum *β*-lactam antibiotics [5]. Nosocomial infections caused by *A. baumannii* are difficult to treat due to multidrug resistance (MDR), which severely limits the options for therapeutic treatment due to increased resistance to carbapenems, which are used as a last resort for antibiotic therapy to eliminate infections caused by multidrug-resistant Gram-negative bacteria [6]. When bacteria are responsible for antibiotic resistance to at least one antimicrobial agent from three or more classes of antimicrobials (for example, penicillins, aminoglycosides, cephalosporins, fluoroquinolones, or tetracyclines), they are considered to be multidrug resistant [7]. Antibiotic resistance in *A*. *baumannii* occurs due to the enzymatic degradation of antibiotics, mutations/modification of target sites, reduced expression of porins, and overexpression of multidrug efflux pumps [8]. Resistance to carbapenems is often mediated by *β*-lactamases including extended-spectrum *β*-lactamases (ESBLs) and AmpC [9]. ESBLs conferring resistance to broad-spectrum cephalosporins and carbapenemases conferring resistance to carbapenems are the greatest concern. An ESBL and an AmpC *β*-lactamase in a single isolate confer resistance to carbapenems that are usually the drug of choice in the treatment of *A. baumannii* infection [10]. Due to resistance to multiple antibiotics, *A. baumannii* is associated with high mortality [11]. In 2017, this pathogen was included in the World Health Organization (WHO) global priority list of drug-resistant bacteria in order to highlight the need for research development and the urgent need for new antibiotics [12]. However, clinical data evaluating potentially effective treatment methods are currently insufficient, and randomized clinical trials have not shown that a single antimicrobial agent or combination therapy is more efficient [13]. Therefore, it is very important to maintain the effectiveness of already available antibiotics and to pay considerable attention to the prevention of the spread of *A. baumannii* infection in healthcare facilities [13,14].

According to the European Antimicrobial Resistance Surveillance Network (EARS-Net) report, the resistance rate of *Acinetobacter* spp. to carbapenems in Lithuania in 2021 was 96.1% [15]. The aim of this study was to determine the characteristics of resistance of *A. baumannii* strains to different antibiotics, to evaluate the associations between the phenotypes of resistance to *β*-lactams and other tested antibiotics, and to evaluate the changes in antibiotic resistance of *A. baumannii* strains after 5 years by comparing the periods of 2016–2017 and 2020–2021.

## 2. Materials and Methods

### 2.1. Bacterial Strains

*A*. *baumannii* strains were isolated from different clinical specimens of patients treated in the Hospital of Lithuanian University of Health Sciences. Isolates were defined as MDR if they were resistant to at least one antimicrobial agent from three or more classes of antimicrobials: penicillins (ampicillin/sulbactam, piperacillin/tazobactam) or cephalosporins (ceftazidime, cefepime), fluoroquinolones (ciprofloxacin), and aminoglycosides (amikacin, tobramycin, gentamicin). Specimens were obtained from wounds, biopsy, bronchial secretions, blood, sputum, pus, abdominal fluid, implants, pleural fluid, urine, and cerebrospinal fluid of hospitalized patients. All the isolates were cultivated on blood agar and MacConkey agar for 17–24 h, and pure cultures were identified using a MALDI-TOF MS mass spectrometer (Brucker Daltonics Gmbh, Bremen, Germany).

### 2.2. Antimicrobial Susceptibility Testing

Antimicrobial susceptibility testing was performed by a disk diffusion method on Müller–Hinton agar (MH II according to EUCAST, Graso Biotech Microbiology Systems, Owidz, Poland). All inoculated plates were incubated for a total of 16–20 h at 35 °C ± 1 °C in an ambient air incubator after inoculation with organisms and placement of disks. All the strains were tested for sensitivity to the following antibiotics: ceftazidime, cefepime, gentamicin, amikacin, ciprofloxacin, ampicillin/sulbactam, piperacillin/tazobactam, imipenem, meropenem, doxycycline, tigecycline, tetracycline, and sulfamethoxazole/trimethoprim disks (Becton Dickinson Microbiology Systems) were used. The diameter of the inhibition zone was measured in millimeters using a ruler. Inhibition zone diameters were interpreted according to the European Committee on Antimicrobial Susceptibility Testing (EUCAST) recommendations [16]. The results were interpreted according to the EUCAST breakpoints. All isolated and identified *A. baumannii* strains were frozen at −80 °C.

### 2.3. Phenotypic Methods for the Determination of ESBL Type

A total of 233 *A. baumannii* strains resistant to carbapenems (imipenem and/or meropenem) were tested using a four-combination disk test (CDT) to determine the production of ESBL and AmpC (Abtek Biologicals, Liverpool, UK, in 2016 and 2017; Liofilchem^®^, Roseto degli Abruzzi, Italy, in 2021 and 2022). Bacterial suspensions (0.5 McFarland) of *A. baumannii* strains were thawed at room temperature for about 15 min and cultured on MH agar using different swabs. With all aseptic precautions, the following antibiotic disks were placed on inoculated MH agar plates: 30 μg cefotaxime (CTX 30), 30 μg cefotaxime plus 10 μg clavulanic acid (CTL 40), 30 μg cefotaxime plus 200 μg cloxacillin (CTC 230), 30 μg cefotaxime plus 200 μg cloxacillin plus 10 μg clavulanic acid (CTLC 240). The plates were incubated at 37 °C for 18–24 h in an ambient-air incubator. An increase of ≥5 mm in the zone diameter with the CTL disk as compared with the CTX disk alone indicated ESBL production. The isolate was defined as an AmpC producer when the following was observed: (a) there was an increase of ≥5 mm in the zone diameter with the CTC 230 disk as compared to the CTX 30 disk or (b) there was an increase of ≥5 mm in the zone diameter with the CTLC 240 disk as compared to the CTL 40 disk or (c) there was an increase in the diameter with the CTLC 240 disk measuring < 5 mm as compared to the CTC 230 disk.

*A. baumannii* strains were also considered ESBL and AmpC producers if the diameter of the CTL 40 disk was smaller than ≤5 mm as compared to the CTX 30 disk, but the diameter of the CTLC 240 disk was greater than ≥5 mm as compared to the CTL 40 disk and the diameter of the CTLC 240 disk was greater than ≥5 mm as compared to CTC 230 disk. If the diameters of the disks differed from each other by 2 mm or less, the strain being tested was neither an ESBL nor an AmpC producer.

All *A. baumannii* strains resistant to carbapenems (imipenem and/or meropenem) were tested using a combination meropenem disk test. The advantage of the test is that it discriminates between carbapenem-susceptible, KPC-producing, metallo *β*-lactamase (MBL)-producing, and double carbapenemase–producing bacteria. Bacterial suspensions (0.5 McFarland) of *A. baumannii* were made and cultured on Mueller–Hinton agar using different swabs, and then four meropenem disks were placed. EDTA (0.1 M, 10 μL) was added on the second disk; 20 μL of phenylboronic acid (20 g/L) on the third disk; and 20 μL of phenylboronic acid (20 g/L) plus 10 μL EDTA (0.1 M) on the fourth disk. The plates were incubated at 37 °C for 18–24 h in an ambient-air incubator. Interpretation of the results of the combination meropenem disk test was based on the comparison among the inhibition zones of four meropenem disks. If no carbapenemase was present, the diameter of the disk with added inhibitors was similar in size (≥5 mm) compared to the diameter of the meropenem disk alone. If an isolate was a KPC producer, there was an increase of ≥5 mm in the diameters of the disks supplemented with phenylboronic acid as compared to the disks without phenylboronic acid. MBL production was confirmed by an increase of ≥5 mm in the diameters of the disks that were supplemented with EDTA. In the case of a KPC plus MBL producer, the fourth disk had the largest zone diameter of all. EDTA- and phenylboronic acid–supplemented disks were ≥5 mm larger in diameter than the meropenem disk alone. Figure 1 shows the workflow diagram.

The Kaunas Regional Biomedical Research Ethics Committee approved this study (No. BE10-0016).

### 2.4. Statistical Analysis

The chi-square (χ^2^) criterion was used for the comparison of categorical data, and continuous data were compared with Student’s t test. The results were considered statistically significant at *p* < 0.05. Statistical package SPSS 27.0 for Windows was used for the data analysis.

## 3. Results

A total of 233 *A*. *baumannii* strains were isolated from different clinical specimens of patients treated in the Hospital of Lithuanian University of Health Sciences: 130 in 2016–2017 and 103 in 2021–2022. In 2016–2017, 44 (33.8%) isolates were collected from women and 86 (66.2%) from men, with a mean age of 62.9 (SD 17.5) years. In 2021–2022, 29 (28.2%) isolates were collected from women and 74 (71.8%) from men, with a mean age of 62.7 (SD 14.4) years.

In 2016–2017 and 2021–2022, the highest number of *A. baumannii strains* were isolated from patients hospitalized in intensive care units (67.7%, *n* = 88 and 59.2%, *n* = 61, respectively), followed by surgical wards (22.3%, *n* = 29 and 29.1%, *n* = 30, respectively), and medical wards (10.0%, *n* = 13 and 11.7%, *n* = 12, respectively). *A. baumannii* was predominantly isolated from bronchial secretions (60.1%, *n* = 140, respectively), followed by urine (10.3%, *n* = 24, respectively), and wounds and pus (8.6%, *n* = 20, respectively). Table 1 shows more detailed information on the sources for the isolation of *A. baumannii* strains of the two different periods.

Examination of all *A. baumannii* strains and determination of their sensitivity to antibiotics showed that all *A. baumannii* strains were resistant to ciprofloxacin, and more than 80% were resistant to carbapenems, piperacillin/tazobactam, gentamicin, and tobramycin.

We compared the resistance rates of *A. baumannii* strains to antibiotics in 2016–2017 and in 2021–2022. The percentage of doxycycline-resistant *A. baumannii* strains almost tripled from 2016–2017 to 2021–2022 (*p* < 0.001). Moreover, the percentages of tetracycline- and tigecycline-resistant strains were significantly greater in 2016–2017 as compared to 2021–2022 (both *p* < 0.001). In contrast, the percentage of *A. baumannii* strains resistant to trimethoprim/sulfamethoxazole was significantly lower in 2021–2022 as compared with 2016–2017 (*p* < 0.001). Resistance rates to all tested antibiotics of the two periods are shown in Table 2.

Of all *A. baumannii* strains tested, 9.9% (*n* = 23) did not produce any *β*-lactamase. More than one-third of *A. baumannii* strains (34.8%, *n* = 81) produced only one type of *β*-lactamases, i.e., AmpC or KPC or ESBL; 51.5% (*n* = 120) produced two different types of *β*-lactamases, i.e., AmpC plus KPC, AmpC plus ESBL, or ESBL plus KPC; and 3.8% (*n* = 9) produced all three types of *β*-lactamases (Figure 2).

In 2016–2017, 17.7% (*n* = 23) of *A. baumannii* strains did not produce any *β*-lactamase, whereas in 2021–2022, there were no such *A. baumannii* strains (*p* < 0.001). In 2016–2017, 62.3% of all tested strains produced only one type of *β*-lactamase, whereas in 2021–2022, no such strains were detected (*p* < 0.001). The percentage of isolates producing two different types of *β*-lactamases was significantly greater in 2021–2022 than in 2016–2017 (*p* < 0.001). Only 2.3% (*n* = 3) of *A. baumannii* isolates produced all three types *β*-lactamases in 2016–2017, whereas in 2021–2022, this percentage accounted for 5.8% (*n* = 6) (*p* < 0.001) (Figure 3).

### Assessment of Antibiotic Resistance Based on β-Lactamase Type

Resistance rates of *A. baumannii* strains producing and not producing AmpC, ESBL, and KPC in response to all antibiotics investigated are shown in Table 3. AmpC-producing *A. baumannii* strains were more frequently resistant to tigecycline, tetracycline, and doxycycline than those strains not producing AmpC (*p* < 0.001 each). However, AmpC-producing strains were less frequently resistant to gentamicin (*p* < 0.001), tobramycin (*p* = 0.031), amikacin (*p* < 0.001), and trimethoprim/sulfamethoxazole (*p* < 0.001) as compared with AmpC-non-producing strains. Significant differences in the resistance rates between ESBL-producing and ESBL-non-producing *A. baumannii* strains were noted only for two groups of antimicrobial drugs—ampicillin/sulbactam (*p* = 0.014) and tigecycline (*p* = 0.021)—with the resistance rates being greater in ESBL-non-producing strains. The percentages of KPC-producing strains resistant to piperacillin/tazobactam (*p* = 0.012), carbapenems (*p* = 0.040), tigecycline (*p* < 0.001), tetracycline (*p* < 0.001), and doxycycline (*p* < 0.001) were significantly greater than those not producing KPC. The resistance rate of KPC-producing strains to trimethoprim/sulfamethoxazole was significantly lower than that of KPC-non-producing strains (*p* = 0.038).

Antibiotic resistance rates of *A. baumannii* strains producing different types and numbers of *β*-lactamases are presented in Table 4. Isolates producing one and two types of *β*-lactamase were more often resistant to ampicillin/sulbactam as compared to all three types of *β*-lactamase-producing strains (87.7% and 85.0% vs. 55.6%, *p* = 0.031). *A. baumannii* strains producing one type of *β*-lactamase were more often resistant to trimethoprim/sulfamethoxazole as compared with strains producing two or three types of *β*-lactamases (97.5% vs. 70.0% and 77.8%, *p* < 0.001). *A. baumannii* strains producing two or three types of *β*-lactamases were more often resistant to tigecycline as compared with strains producing one type of *β*-lactamase (87.5% and 77.8% vs. 67.9%, *p* < 0.001), tetracycline (94.2% and 88.4% vs. 64.2%, *p* < 0.001), and doxycycline (88.3% and 88.9% vs. 25.9%, *p* < 0.001) as compared with strains producing one type of *β*-lactamase.

## 4. Discussion

The number of cases of bacterial nosocomial infections resistant to antimicrobial agents has increased worldwide over the last decade [4]. *A. baumannii* is considered one of the most important causes of antimicrobial resistance in health care facilities. *A. baumannii* strains cause outbreaks around the world due to their remarkable ability to adapt to the changes in the environment and to acquire different resistance mechanisms against multiple antimicrobials [4]. This study aimed to determine the types of *β*-lactamases and their combinations produced by MDR *A. baumannii* strains. It also aimed to evaluate associations between antibiotic resistance and produced types of *β*-lactamases and to evaluate the changes in antibiotic resistance during the periods of 2016–2017 and 2020–2021.

According to the 2017 European Antimicrobial Resistance Surveillance Network (EARS-Net) report, the proportions of carbapenem-resistant *Acinetobacter* spp. collected from invasive infections were particularly high in many southern and eastern European countries, where resistance proportions often exceeded 50% (such as 95% in Greece, 79% in Italy, and 53% in Hungary). The EARS-Net analysis also showed that, on average, more than  70% of carbapenem-resistant *Acinetobacter* spp. were also resistant to ciprofloxacin or gentamicin in southern and eastern Europe [17]. In our study, all *A. baumannii* strains were resistant to ciprofloxacin, and resistance rates of *A. baumannii* strains to carbapenems, piperacillin/tazobactam, gentamicin, and tobramycin were high (more than 80%), supporting the findings of the EARS-Net report.

In our study, only 11.6% (*n* = 27) of all *A. baumannii* isolates were ESBL producers. However, we did not investigate to which particular *β*-lactamase class the investigated ESBLs belonged. The study by Kaur et al., investigated the prevalence of ESBLs in *A. baumannii* isolates obtained from various clinical samples of one Indian institution and reported that of the 116 *A. baumannii* isolates, ESBL production was documented in 32 isolates (27.5%) [18]. Another study carried out in one teaching and referral Ethiopian hospital showed that ESBL production was observed in 55.8% (*n* = 24) of the 38 *A. baumannii* isolates collected from different wards such as the ICU, delivery room, and operating room [19]. In both studies, the ESBL production by *A. baumannii* strains was greater as compared with that in our study.

In the present study, 60.9% of isolates were AmpC producers as determined by the CDT method. Other authors also reported a similar percentage (64.63%) of AmpC producers among 82 *Acinetobacter* spp. isolates using the same method [20]. With regard to KPC producers among *A. baumannii* isolates, we found that this percentage was 76.8%. Abouelfetouh et al. [21] investigated carbapenemase production in 74 carbapenem-resistant *A. baumannii* strains isolated from different clinical specimens of one Egyptian university hospital by various phenotypic methods and determined that 79.7% (*n* = 59) of the isolates were carbapenemase producers by the CDT method.

We showed that KPC-producing *A. baumannii* strains were more frequently resistant to carbapenems, piperacillin/tazobactam, and tetracyclines. The KPC enzyme hydrolyzes *β*-lactam antibiotics, including carbapenems [22], and this has been associated with *A. baumannii* isolates being more frequently resistant to *β*-lactam antibiotics, including carbapenems. The resistance rate of *A. baumannii* to fluoroquinolones, aminoglycosides, and trimethoprim/sulfamethoxazole was also high in our study. MDR *A. baumannii* strains remain susceptible to only a few antibiotics, such as minocycline/tigecycline and polymyxins [23]. Monotherapy or combination therapy with next-generation tetracycline class antibiotics (e.g., tigecycline) is often employed as a last-resort measure to treat MDR and XDR *A. baumannii* infections [24]; however, our study shows that treatment with antibiotics of the tetracycline class leads to greater resistance, and these antibiotics lose their effectiveness against infections caused by *A. baumannii*.

Compared with other studies, we found that *A. baumannii* strains producing two different types of *β*-lactamases—KPC and AmpC—were detected more frequently (46.3%), whereas the studies by Hans et al. [25] and Das and Basak [26], both conducted in India, reported these percentages being 10% and 23.3%, respectively. In our study, all three types of *β*-lactamases—AmpC, KPC, and ESBL—were produced by 3.8% of isolates. According to the results of other research, carried out in in a tertiary care hospital of South India, 1.78% of the isolates produced all three types of *β*-lactamases [27].

Comparison of two different periods revealed that the production of different types of *β*-lactamases was documented more frequently in 2020–2021 than 2016–2017. This is a serious warning, as *β*-lactamases pose a significant threat to the effectiveness of antibiotics currently available for medical use. Other literature resources have also shown that *β*-lactamases are important in the emergence of antimicrobial-resistant strains and have reported a large increase in resistance among *A. baumannii* strains in healthcare settings [28,29]. Thus, due to such a high prevalence of resistance, antibiotics must be used judiciously by clinicians, and appropriate infection control measures need to be implemented to control the spread of infections in hospitals.

## 5. Conclusions

An increasing body of evidence suggests a worldwide loss of critically important antimicrobial medicines and emphasizes the need to protect currently available antibiotics and develop new ones. The present study also shows a warning trend toward greater resistance to the antibiotics of the tetracycline class, as the resistance rates to ampicillin/sulbactam, tigecycline, tetracycline, and doxycycline, as well as the frequency of isolation of *A. baumannii* strains producing two different types of *β*-lactamases (AmpC plus KPC, AmpC plus ESBL or ESBL plus KPC) or three types of *β*-lactamases (AmpC, KPC, and ESBL), were significantly greater in 2020–2021 as compared with 2016–2017.

## Figures and Tables

**Figure 1 medicina-59-01386-f001:**
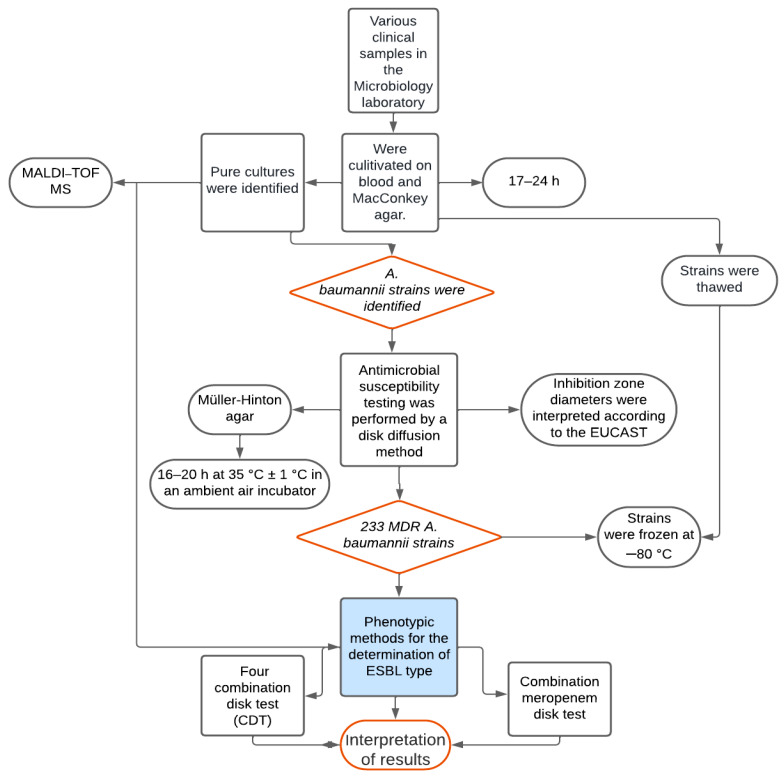
Diagram of the study performed.

**Figure 2 medicina-59-01386-f002:**
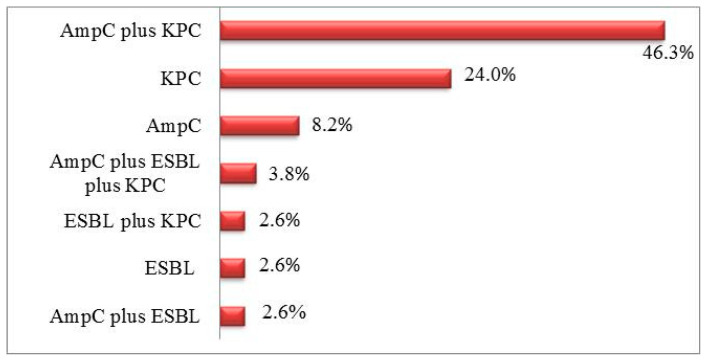
Percentage distribution of *A. baumannii* strains by produced *β*-lactamases and their combinations.

**Figure 3 medicina-59-01386-f003:**
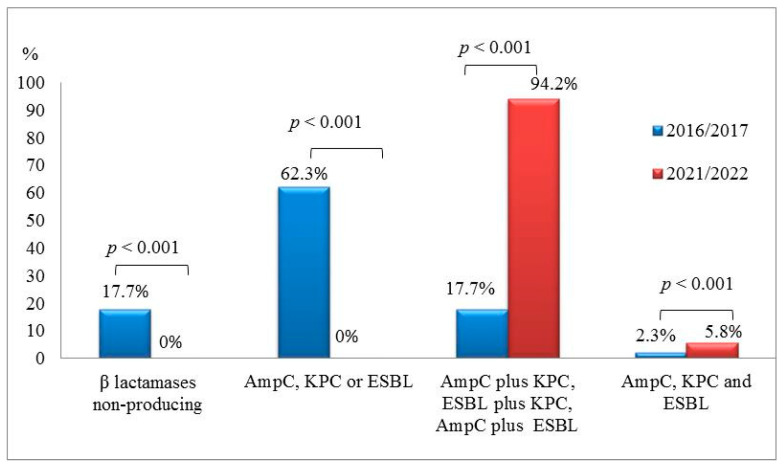
Percentage distribution of *A. baumannii* strains producing different types of *β*-lactamases or their combinations in 2016–2017 and 2021–2022.

**Table 1 medicina-59-01386-t001:** Distribution of *Acinetobacter baumannii* strains by isolation sources and two different periods.

Source	Total N = 233 *n* (%)	2016–2017 N = 130 *n* (%)	2021–2022 N = 103 *n* (%)
Bronchial secretions	140 (60.1)	88 (37.8)	52 (22.3)
Urine	24 (10.3)	12 (5.2)	12 (5.2)
Wounds and pus	20 (8.6)	9 (3.8)	11 (4.8)
Abdominal fluid and bile	19 (8.2)	8 (3.4)	11 (4.8)
Blood	16 (6.9)	5 (2.1)	11 (4.8)
Sputum	5 (2.1)	2 (0.9)	3 (1.2)
Biopsy	4 (1.7)	3 (1.3)	1 (0.4)
Cerebrospinal fluid	3 (1.3)	1 (0.4)	2 (0.9)
Pleural fluid	1 (0.4)	1 (0.4)	0 (0)
Implant	1 (0.4)	1 (0.4)	0 (0)

**Table 2 medicina-59-01386-t002:** Resistance of *A. baumannii* strains to different antibiotics comparing 2016–2017 and 2021–2022.

Antimicrobial Drug	2016–2017 N = 130 *n* (%)	2021–2022 N = 103 *n* (%)	*p*
Ampicillin/sulbactam	104 (80.0)	96 (93.2)	0.004
Piperacillin/tazobactam	127 (97.7)	103 (100.0)	0.257
Ceftazidime	129 (99.2)	102 (99.0)	1.0
Cefepime	124 (95.4)	100 (97.1)	0.735
Imipenem	127 (97.7)	102 (99.0)	0.632
Meropenem	127 (97.7)	102 (99.0)	0.632
Ciprofloxacin	130 (100.0)	103 (100.0)	–
Gentamicin	111 (85.4)	89 (86.4)	0.824
Tobramycin	124 (95.4)	93 (90.3)	0.127
Amikacin	105 (80.8)	87 (84.5)	0.462
Tigecycline	80 (61.5)	99 (96.1)	<0.001
Tetracycline	81 (62.3)	103 (100.0)	<0.001
Doxycycline	42 (32.3)	99 (96.1)	<0.001
Trimethoprim/sulfamethoxazole	124 (95.4)	69 (67.0)	<0.001

**Table 3 medicina-59-01386-t003:** Comparison of antimicrobial resistance between *A. baumannii* strains producing and not producing only one type of *β*-lactamase (AmpC, ESBL, and KPC).

Antimicrobial Drug	AmpC-Producing N = 142 *n* (%)	AmpC-Non-Producing N = 91 *n* (%)	*p*	ESBL-Producing N = 27 *n* (%)	ESBL-Non-Producing N = 206 *n* (%)	*p*	KPC-Producing N = 179 *n* (%)	KPC-Non-Producing N = 54 *n* (%)	*p*
Ampicillin/sulbactam	117 (82.4)	83 (91.2)	0.600	19 (70.4)	181 (87.9)	0.014	154 (86.0)	46 (85.2)	0.875
Piperacillin/tazobactam	139 (97.9)	91 (100.0)	0.163	26 (96.3)	204 (99.0)	0.236	179 (100.0)	51 (94.4)	0.012
Ceftazidime	141 (99.3)	90 (98.9)	0.750	26 (96.3)	205 (99.5)	0.219	178 (99.4)	53 (98.1)	0.411
Cefepime	135 (95.1)	89 (97.8)	0.291	24 (88.9)	200 (97.1)	0.219	173 (96.6)	51 (94.4)	0.461
Imipenem	138 (97.2)	91 (100.0)	0.106	26 (96.3)	203 (98.5)	0.391	178 (99.4)	51 (94.4)	0.040
Meropenem	138 (97.2)	91 (100.0)	0.106	26 (96.3)	203 (98.5)	0.391	178 (99.4)	51 (94.4)	0.040
Ciprofloxacin	142 (100.0)	91 (100.0)	–	27 (100.0)	206 (100.0)	–	179 (100.0)	54 (100.0)	–
Gentamicin	113 (79.6)	87 (95.6)	<0.001	21 (77.8)	179 (86.9)	0.201	156 (87.2)	44 (81.5)	0.295
Tobramycin	128 (90.1)	89 (97.8)	0.031	27 (100.0)	190 (92.2)	0.133	165 (92.2)	52 (96.3)	0.294
Amikacin	107 (75.4)	85 (93.4)	<0.001	20 (74.1)	172 (83.5)	0.279	151 (84.4)	41 (75.9)	0.159
Tigecycline	122 (85.9)	57 (62.6)	<0.001	16 (59.3)	163 (79.1)	0.021	148 (82.7)	31 (57.4)	<0.001
Tetracycline	129 (90.8)	55 (60.4)	<0.001	21 (77.8)	163 (79.1)	0.872	152 (84.9)	32 (59.3)	<0.001
Doxycycline	122 (85.9)	19 (20.9)	<0.001	16 (59.3)	125 (60.7)	0.887	119 (66.5)	22 (40.7)	0.001
Trimethoprim/ Sulfamethoxazole	103 (72.5)	90 (98.9)	<0.001	22 (81.5)	171 (83.0)	0.790	143 (79.9)	50 (92.6)	0.038

**Table 4 medicina-59-01386-t004:** Comparison of antimicrobial resistance among *A. baumannii* strains producing different types and numbers of *β*-lactamases.

Antimicrobial Drug	Producing only One Type of *β*-Lactamase AmpC or KPC or ESBL N = 81 *n* (%)	Producing Two Different Types of *β*-Lactamases AmpC Plus KPC, AmpC Plus ESBL or ESBL Plus KPC N = 120 *n* (%)	All Three Types of *β*-Lactamases AmpC, KPC and ESBL N = 9 *n* (%)	*p*
Ampicillin/sulbactam	71 (87.7) ^a^	102 (85.0) ^b^	5 (55.6) ^c^	0.031 (^a,b^ vs. ^c^)
Piperacillin/tazobactam	79 (97.5)	119 (99.2)	9 (100)	0.682
Ceftazidime	80 (98.8)	119 (99.2)	9 (100)	0.938
Cefepime	78 (96.3)	115 (95.8)	8 (88.9)	0.526
Imipenem	79 (97.5)	118 (98.3)	9 (100)	0.842
Meropenem	79 (97.5)	118 (98.3)	9 (100)	0.842
Ciprofloxacin	81 (100.0)	120 (100.0)	9 (100.0)	–
Gentamicin	75 (92.6)	97 (80.8)	7 (77.8)	0.824
Tobramycin	77 (95.1)	108 (90.0)	9 (100)	0.197
Amikacin	71 (87.7)	93 (77.5)	7 (77.8)	0.087
Tigecycline	55 (67.9) ^a^	105 (87.5) ^b^	7 (77.8) ^c^	<0.001 (^a^ vs. ^b,c^)
Tetracycline	52 (64.2) ^a^	113 (94.2) ^b^	8 (88.9) ^c^	<0.001 (^a^ vs. ^b,c^)
Doxycycline	21 (25.9) ^a^	106 (88.3) ^b^	8 (88.9) ^c^	<0.001 (^a^ vs. ^b,c^)
Trimethoprim/ Sulfamethoxazole	79 (97.5) ^a^	84 (70) ^b^	7 (77.8) ^c^	<0.001 (^a^ vs. ^b,c^)

Strains producing one type of β-lactamase were more often resistant to antibiotics indicated by letter ^a^, Strains producing two, three types of β-lactamase were more often resistant to antibiotics indicated by letter ^b^, Strains producing three types of β-lactamase were more often resistant to antibiotics indicated by letter ^c^.

## Data Availability

The data presented in this study are available on request from the corresponding author.

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
