# Peer review of "Associations between β-Lactamase Types of Acinetobacter baumannii and Antimicrobial Resistance"

_medicina, 2023, doi:10.3390/medicina59081386_

Round 1

Reviewer 1 Report

1.  The author has explained the ESBL, but the specific class provides more informative ie., they should explain the class(A-D) in the introduction section.  The classD was well explained earlier with respect to the experimental data.  Hence, authors are advised to refer the following papers 

paper -1: https://doi.org/10.3389/fchem.2023.1090630

paper -2: https://doi.org/10.1099/jmm.0.001233

2.  The authors are advised to check the following sentence:

We compared the resistance rates of A. baumannii strains to antibiotics in 2016–2017 and 2021–2022. Resistance of A. baumannii strains to doxycycline from 2016–2017 to 2021–2022 increased the most (page number 4)

3.  The author had demonstrated the study appropriately. Still it requires some analysis.  They started with the different strains of producing beta-lactamases and tested against with each antibiotics.  Also, the 2020-21 is higher than 2016-17.  The author has to describe which class of beta-lactamases is produced (for ex: ClassA or any).  In the earlier studies, the report was given as Class-D.  Hence, author has to work on to identify which class suppose to be.  In addition to this they didnt mention the strains here.

4.  Author described that carbapenems (meropenem/imipenem) resistance to the beta-lactamase.  Still, the author has to refer the above-mentioned articles to get clarity by describing how it correlates. 

5.  re-write this sentence:

Monotherapy or combination therapy with next-generation tetracycline class antibiotics (e.g., tigecycline) is often employed as a last-resort measure to treat MDR and XDR A. baumannii infections (20); however, our study suggests that antibiotics of the tetracycline class are losing their effectiveness against infections caused by A. baumannii.

Moderate improvement of English language is required

Author Response

Responses to Reviewer 5 Comments

First at all, we would like to thank the Reviewer for his/her time and efforts put into reviewing our manuscript. All the remarks given by the reviewer and all the corrections done in the manuscript are highlighted in a red color. We also would like to point out that four different reviewers reviewed our manuscript and therefore we tried to make optimal corrections taking the remarks by all reviewers into consideration.

Remark 1: The author has explained the ESBL, but the specific class provides more informative ie., they should explain the class (A-D) in the introduction section. The class D was well explained earlier with respect to the experimental data. Hence, authors are advised to refer the following papers

paper -1: https://doi.org/10.3389/fchem.2023.1090630

paper -2: https://doi.org/10.1099/jmm.0.001233

In our study, we investigated the ESBL without determining the particular class (D or B). Therefore, no attention is given to the classes in the introduction, and it would be difficult to refer to the articles mentioned above.

Remark 2: The authors are advised to check the following sentence:

We compared the resistance rates of A. baumannii strains to antibiotics in 2016–2017 and 2021–2022. Resistance of A. baumannii strains to doxycycline from 2016–2017 to 2021–2022 increased the most (page number 4)

We revised this sentence and now it appears to be as follows:

“Resistance of A. baumannii strains to doxycycline from 2016–2017 to 2021–2022 increased the most, i.e. the percentage of doxycycline-resistant A. baumannii strains almost tripled (p < 0.001). Meanwhile, the percentages of tetracycline- and tigecycline-resistant strains showed a 38%-point and a 35%-point increase, respectively, as 2016–2017 was compared with 2021–2022 (both p < 0.001).”

Remark 3:

The author had demonstrated the study appropriately. Still it requires some analysis. They started with the different strains of producing beta-lactamases and tested against with each antibiotics. Also, the 2020-21 is higher than 2016-17. The author has to describe which class of beta-lactamases is produced (for ex: ClassA or any). In the earlier studies, the report was given as Class-D. Hence, author has to work on to identify which class suppose to be. In addition to this they didn’t mention the strains here,

During this study, we performed phenotyping analysis that does not allow determining to which classes ESBLs belong to. This was additionally mentioned in the discussion and could be considered as a limitation.

Remark 4:

Author described that carbapenems (meropenem/imipenem) resistance to the beta-lactamase. Still, the author has to refer the above-mentioned articles to get clarity by describing how it correlates.

We find it difficult to refer to and cite the above-mentioned articles as we employed different methods in our study.

 Remark 5:

re-write this sentence: “Monotherapy or combination therapy with next-generation tetracycline class antibiotics (e.g., tigecycline) is often employed as a last-resort measure to treat MDR and XDR A. baumannii infections (20); however, our study suggests that antibiotics of the tetracycline class are losing their effectiveness against infections caused by A. baumannii.”

 The sentence was rewritten as follows:

Monotherapy or combination therapy with next-generation tetracycline class antibiotics (e.g., tigecycline) is often employed as a last-resort measure to treat MDR and XDR A. baumannii infections (20); however, our study shows that treatment with antibiotics of the tetracycline class leads to increased resistance and these antibiotics lose their effectiveness against infections caused by A. baumannii.”

Reviewer 2 Report

The manuscript has presented multiple flaws in either the form and the content. Even the content in the content could be accepted (after revisions), the form of the manuscript cannot be acceptable for a submitted manuscript for a such a journal. This demonstrate the lack of rigor and seriousness and the manuscript

1.  The font types and sizes are different from a paragraph to another 

2. The English language contains multiple errors requiring an extensive revision of the manuscript.

3. The manuscript contains also multiple spelling errors (gram, by Dutch microbiologist..)

4. The authors includes Klebsiella as a key word?

5. The authors used multiple "orphan sentences " without any reference (expl: multiple sentences in page 2). They also used generalized ideas (for all bacteria) and limited it just to AB (Ex: When A. baumannii is responsible…resistant (4).).

6. The authors used practically the same sentences word by word in the manuscript and in the abstract (exp: all A. baumannii strains were resistant to ciprofloxacin; more than 80% were…) and also in the introduction and the methods (exp: "isolate were defined as MDR…..and gentamycin". This should be deleted from the methods.

7. the authors should add a reference for the molecular characterization

8. The results should be extensively revised and avoid to compare every time between the two periods.

They should also begin with the origin of the strains (the total and for each period) and add a table to support the data.

This part represents my main concern. How did the authors choose these two periods ? Why they did not include the period 2018-2020. They should justify this choice.

They should also understand that they are not studying the evolution of AB antibiotic resistance but just comparing the two periods (they should avoid to use the terms, changes, evolution, increase, decrease..) this needs to study all years to talk about the evolution.

In addition, the results are poorly organized and are repetitive. You should begin with the ATBR, then AMR (multi-resistance). They should also begin with the frequency of each ESBL and then reports the frequency of the associations.

Tables 3, 4, and 5 could be associated in just one table to avoid repetition and to facilitate the comprehension. The future table should be improved (exp what the asterisk mean?).

The discussion should be revised and the first sentences should be referenced.(exp: The study by Kaur et al. reported (13) that of the…." , you should also explain where reference 13 and 14 were conducted).

The conclusion is very simplistic and did not provide the most important findings. Revise

Conclusion: In conclusion, the manuscript is f very poor quality and is not yet ready not only for publication but for submission.

Extensive editing of English language required

Author Response

Response to Reviewer 8 Comments

First at all, we would like to thank the Reviewer for his/her time and efforts put into reviewing our manuscript. All the remarks given by the reviewer and all the corrections done in the manuscript are highlighted in a red color. We also would like to point out that four different reviewers reviewed our manuscript and therefore we tried to make optimal corrections taking the remarks by all reviewers into consideration.

Remark 1: The font types and sizes are different from a paragraph to another 

We apologize for these inconsistencies regarding formatting. It seems to be caused by the conversion of a Word file to the pdf one. The revised manuscript is consistent regarding the font and size.

Remark 2: The English language contains multiple errors requiring an extensive revision of the manuscript.

We appreciate the reviewer for this comment. Despite the manuscript was edited by a professional English language editor, there is a possibility for some errors to be left. It would be very useful if the reviewer could point out other multiple errors present in our manuscript. The resubmitted manuscript was again reviewer by the same editor. If the reviewer still sees multiple errors, we would be grateful for indicating them.

Remark 3: The manuscript contains also multiple spelling errors (gram, by Dutch microbiologist..)

If the reviewer had in his/her mind that word “gram” in the phrases such as “gram-negative” or gram-positive” should be typed from the capital letter, we would like to gently disagree. If we had referred to Gram staining or Gram as a biologist, then word “Gram” would have been capitalized. However, phrases “gram-negative” or gram-positive” should be lowercase as it is indicated by multiple medical dictionaries and the information on the CDC website:   

https://wwwnc.cdc.gov/eid/page/preferred-usage#:~:text=Gram%20should%20be%20capitalized%20and,used%20as%20a%20unit%20modifier.

Considering the phrase “by Dutch microbiologist” indicated by the reviewer, we guess that the reviewer pointed out the missing indefinite article. However, as this phrase is followed by the name and surname of this microbiologist, no any article is needed. Please compare the following examples:

 “by Dutch microbiologist Martinus Willem Beigerinck” and

“by a Dutch microbiologist by the name of Martinus Willem Beigerinck”

Remark 4: The authors includes Klebsiella as a key word?

Actually, the whole keyword was “Klebsiella pneumoniae carbapenemase”, but following the reviewer’s remark, this keyword was removed.

Remark 5: The authors used multiple "orphan sentences" without any reference (expl: multiple sentences in page 2). They also used generalized ideas (for all bacteria) and limited it just to AB (Ex: When A. baumannii is responsible…resistant (4).).

The shortcomings noted by the reviewer were corrected.

Remark 6: The authors used practically the same sentences word by word in the manuscript and in the abstract (exp: all A. baumannii strains were resistant to ciprofloxacin; more than 80% were…) and also in the introduction and the methods (exp: "isolate were defined as MDR…..and gentamycin". This should be deleted from the methods.

Actually, no one guide for scientific writing mentions that repeating the same sentences in different parts of a manuscript is not allowable. However, following the reviewer’s suggestion, we rewrote the sentence in the abstract.

Regarding the spelling the word “gentamicin” – it seems that this spelling is also correct. Many websites provide “gentamycin” and “gentamicin” as synonyms. Please follow the different resources, one of them being as follows: https://www.merriam-webster.com/dictionary/gentamicin#:~:text=%E2%80%9CGentamicin.%E2%80%9D%20Merriam%2DWebster,.com%2Fdictionary%2Fgentamicin.

Or https://dailymed.nlm.nih.gov/dailymed/search.cfm?labeltype=all&query=GENTAMICIN

Remark 7: the authors should add a reference for the molecular characterization

The reference has been added.

Remark 8. The results should be extensively revised and avoid to compare every time between the two periods.

We revised the Results section and shortened this part by leaving only the main results.

They should also begin with the origin of the strains (the total and for each period) and add a table to support the data.

The sentences indicating the origin of the strains have been added and the following table has been included.

Table 2. Source of Acinetobacter baumannii isolates in 2016–2017 and 2021–2022.

Samples

2016–2017

N = 130

n (%)

2021–2022

N = 103

n (%)

Total

N = 233

n (%)

Bronchial secretions

88 (37.8)

52 (22.3)

140 (60.1)

Urine

12 (5.2)

12 (5.2)

24 (10.3)

Wounds and pus

9 (3.8)

11 (4.8)

20 (8.6)

Abdominal fluid and bile

8 (3.4)

11 (4.8)

19 (8.2)

Blood

5 (2.1)

11 (4.8)

16 (6.9)

Sputum

2 (0.9)

3 (1.2)

5 (2.1)

Biopsy

3 (1.3)

1 (0.4)

4 (1.7)

Cerebrospinal fluid

1 (0.4)

2 (0.9)

3 (1.3)

Pleural fluid

1 (0.4)

0 (0)

1 (0.4)

Implant

1 (0.4)

0 (0)

1 (0.4)

This part represents my main concern. How did the authors choose these two periods ? Why they did not include the period 2018-2020. They should justify this choice.

They should also understand that they are not studying the evolution of AB antibiotic resistance but just comparing the two periods (they should avoid to use the terms, changes, evolution, increase, decrease..) this needs to study all years to talk about the evolution.

We couldn’t agree more that comparing only two periods (5 years apart) does not allow seeing the whole picture as it would be illustrated by the data provided by a single year.  However, the comparison of only two periods allows technically also seeing the differences after a particular time period. By choosing a 5-year time period, we sought to determine the long-term changes in the resistance of A. baumanii strains to antibiotics with a special attention given to resistance to b-lactams in a hospital setting. Mathematically, the changes seen after a 5-year period could be described as an increase or a decrease and we provide the results by describing them in such a way. The data for each year for this 5-year period were not available.

In addition, the results are poorly organized and are repetitive. You should begin with the ATBR, then AMR (multi-resistance). They should also begin with the frequency of each ESBL and then reports the frequency of the associations.

We minimized the repetition and left the description of the results that were clinically important and statistically significant.

Tables 3, 4, and 5 could be associated in just one table to avoid repetition and to facilitate the comprehension. The future table should be improved (exp what the asterisk mean?). 

Taking the suggestion by the reviewer previous Tables 2–4 were merged. However, Table 5 was left as it as it shows different data regarding the information in it.

Table 3. Comparison of antimicrobial resistance between AmpC-, ESBL-, KPC-producing and AmpC-, ESB-L, KPC-non-producing A. baumannii strains

Antimicrobial drug 

AmpC-producing

N = 142

n (%)

AmpC-non-producing

N = 91

n (%)

P

ESBL-producing

N = 27

n (%)

ESBL-non-producing

N = 206

n (%)

p 

KPC-producing

N = 179

n (%)

KPC-non-producing

N = 54

n (%)

p 

Ampicillin/sulbactam 

117 (82.4)

83 (91.2)

0.600

19 (70.4)

181 (87.9)

0.014

154 (86.0)

46 (85.2)

0.875

Piperacillin/tazobactam 

139 (97.9)

91 (100.0)

0.163

26 (96.3)

204 (99.0)

0.236

179 (100.0)

51 (94.4)

0.012

Ceftazidime 

141 (99.3)

90 (98.9)

0.750

26 (96.3)

205 (99.5)

0.219

178 (99.4)

53 (98.1)

0.411

Cefepime 

135 (95.1)

89 (97.8)

0.291

24 (88.9)

200 (97.1)

0.219

173 (96.6)

51 (94.4)

0.461

Imipenem 

138 (97.2)

91 (100.0)

0.106

26 (96.3)

203 (98.5)

0.391

178 (99.4)

51 (94.4)

0.040

Meropenem 

138 (97.2)

91 (100.0)

0.106

26 (96.3)

203 (98.5)

0.391

178 (99.4)

51 (94.4)

0.040

Ciprofloxacin 

142 (100.0)

91 (100.0)

27 (100.0)

206 (100.0)

179 (100.0)

54 (100.0)

Gentamicin 

113 (79.6)

87 (95.6)

< 0.001

21 (77.8)

179 (86.9)

0.201

156 (87.2)

44 (81.5)

0.295

Tobramycin 

128 (90.1)

89 (97.8)

0.031

27 (100.0)

190 (92.2)

0.133

165 (92.2)

52 (96.3)

0.294

Amikacin 

107 (75.4)

85 (93.4)

< 0.001

20 (74.1)

172 (83.5)

0.279

151 (84.4)

41 (75.9)

0.159

Tigecycline 

122 (85.9)

57 (62.6)

< 0.001

16 (59.3)

163 (79.1)

0.021

148 (82.7)

31 (57.4)

< 0.001

Tetracycline 

129 (90.8)

55 (60.4)

< 0.001

21 (77.8)

163 (79.1)

0.872

152 (84.9)

32 (59.3)

< 0.001

Doxycycline 

122 (85.9)

19 (20.9)

< 0.001

16 (59.3)

125 (60.7)

0.887

119 (66.5)

22 (40.7)

0.001

Trimethoprim/Sulfamethoxazole

103 (72.5)

90 (98.9)

< 0.001

22 (81.5)

171 (83.0)

0.790

143 (79.9)

50 (92.6)

0.038

Regarding the question about the asterisks, we modified this table and instead the asterisks we decided to use the letters to designate what data and how they were compared. Moreover, the last column (where p values are reported) indicates how the data were compared. We hope that this made the comparison easier to understand.

The discussion should be revised and the first sentences should be referenced. (exp: The study by Kaur et al. reported (13) that of the…." , you should also explain where reference 13 and 14 were conducted).

The first sentences in the discussion belong to the same literature source. Therefore, we repeated the citation number. Moreover, we added the information where studies were conducted.

The conclusion is very simplistic and did not provide the most important findings. Revise

The frequency of isolation of A. baumannii strains producing two different types of β-lactamases (AmpC plus KPC, AmpC plus ESBL or ESBL plus KPC) or three types of beta-lactamases (AmpC, KPC, and ESBL) and the resistance rates to ampicillin/sulbactam, tigecycline, tetracycline, and doxycycline increased in 2020–2021 as compared with 2016–2017. The production of two or three types of beta-lactamases by A. baumannii strains was associated with higher resistance rates to tetracyclines.

Reviewer 3 Report

The study titled "Associations between β-lactamase types of Acinetobacter baumannii and antimicrobial resistance" presents a research study focused on the phenomenon of antibiotic resistance in Acinetobacter baumannii strains associated with the hospital sector, specifically focusing on β-lactam resistance. The author discusses the potential changes in antibiotic resistance among the researched bacteria over two time periods, which is a strong aspect of this research. The manuscript is generally well-written and easily understandable. The content is well-structured, providing a clear overview of the study. However, there are several areas of incomprehensibility in the text. Therefore, after a thorough revision of the manuscript, it may be considered for publication. 

Here are the questions and concerns which were the basis for my opinion.

1.     The entire text contains errors regarding font standardization, specifically its size. Please review the correctness of the text formatting according to the journal's provided template, such as citation style. The manuscript should have numbered lines, which would facilitate the review process.

2.     The authors state that they obtained 223 strains, but then specify that they obtained 130 and 103? Please correct this error. Additionally, in my opinion, it is not necessary to include the number of strains obtained in the Materials and Methods section; such information belongs in the Results section. What is important is the "where," "when," and "how."

3.     In my opinion, a more detailed description of the method for obtaining the strains is required. What methods and media were used to obtain pure cultures?

4.     The first sentence ("Of all A. baumannii strains tested...") immediately following Figure 1 should indicate that the data can be found in Figure 1.

5.     The unit representing the values in the table (percentage) should be placed in the table headers, in square brackets.

6.     Discussion: In my opinion, the aim was to determine the antibiotic resistance characteristics of the obtained strains, with particular emphasis on resistance to beta-lactam antibiotics. Additionally, the aim was to determine any associations between resistance phenotypes to beta-lactams and other tested antibiotics. The entire analysis was conducted in two time intervals to assess changes in the resistance structure among the tested strains. Please reconsider the stated objective in the text.

7.     The study identified an increase in resistance among the tested strains, including an expansion of resistance to beta-lactams and an increase in tetracycline resistance cases. What could be the cause of this? Perhaps it would be possible to obtain significantly correlated results between antibiotic consumption in the given region and the increasing antibiotic resistance.

8.     In my opinion, the conclusions extend beyond the actual findings. I advise the authors a reconsideration of the content.

Author Response

Responses to Reviewer 8 Comments

First at all, we would like to thank the Reviewer for his/her time and efforts put into reviewing our manuscript. All the remarks given by the reviewer and all the corrections done in the manuscript are highlighted in a red color. We also would like to point out that four different reviewers reviewed our manuscript and therefore we tried to make optimal corrections taking the remarks by all reviewers into consideration.

 The study titled "Associations between β-lactamase types of Acinetobacter baumannii and antimicrobial resistance" presents a research study focused on the phenomenon of antibiotic resistance in Acinetobacter baumannii strains associated with the hospital sector, specifically focusing on β-lactam resistance. The author discusses the potential changes in antibiotic resistance among the researched bacteria over two time periods, which is a strong aspect of this research. The manuscript is generally well-written and easily understandable. The content is well-structured, providing a clear overview of the study. However, there are several areas of incomprehensibility in the text. Therefore, after a thorough revision of the manuscript, it may be considered for publication. 

Here are the questions and concerns which were the basis for my opinion.

Remark 1. The entire text contains errors regarding font standardization, specifically its size. Please review the correctness of the text formatting according to the journal's provided template, such as citation style. The manuscript should have numbered lines, which would facilitate the review process.

We apologize for these inconsistencies regarding font standardization. It seems to be caused by the conversion of a Word file to the pdf one. Moreover, we have inserted numbered lines despite the template made it difficult to perform.

Remark 2. The authors state that they obtained 223 strains, but then specify that they obtained 130 and 103?  Please correct this error. Additionally, in my opinion, it is not necessary to include the number of strains obtained in the Materials and Methods section; such information belongs in the Results section. What is important is the "where," "when," and "how."

Thank you very much for the remark, the number is corrected. Moreover, the information on the number of strains was removed from Materials and methods.

Remark 3. In my opinion, a more detailed description of the method for obtaining the strains is required. What methods and media were used to obtain pure cultures?

Following the suggestion, a more detailed description was added.

“All the isolates were cultivated on blood agar and MacConkey agar for 17–24 h, and pure cultures were identified using a MALDI-TOF MS mass spectrometer (Brucker Daltonics Gmbh, Germany)”, “Antimicrobial susceptibility testing was performed by a disk diffusion method on Müller-Hinton agar (MH II according to EUCAST, Graso Biotech Microbiology Systems). All inoculated plates were incubated for 16–20 h at 35°C ± 1°C in an ambient air incubator after inoculation with organisms and placement of disks (Becton Dickinson Microbiology Systems)”, and “The diameter of the inhibition zone was measured in millimeters using a ruler. Inhibition zone diameters were interpreted according to the European Committee on Antimicrobial Susceptibility Testing (EUCAST) recommendations [14].”

Remark 4. The first sentence ("Of all A. baumannii strains tested...") immediately following Figure 1 should indicate that the data can be found in Figure 1.

The reference to this figure has been added. After revisions this figure became Fig. 2.

Remark 5. The unit representing the values in the table (percentage) should be placed in the table headers, in square brackets.

Following the reviewer’s suggestion, the units of measure are now indicated in the headers of all tables. Just we used the format of n (%), not square brackets, as square brackets in this case are not applicable. The information “Values are number (percentage)” from the footers of the tables has been removed.

Remark 6. Discussion: In my opinion, the aim was to determine the antibiotic resistance characteristics of the obtained strains, with particular emphasis on resistance to beta-lactam antibiotics. Additionally, the aim was to determine any associations between resistance phenotypes to beta-lactams and other tested antibiotics. The entire analysis was conducted in two time intervals to assess changes in the resistance structure among the tested strains. Please reconsider the stated objective in the text.

The aims in the abstract and at the end of the introduction have been reconsidered and unified.

Remark 7. The study identified an increase in resistance among the tested strains, including an expansion of resistance to beta-lactams and an increase in tetracycline resistance cases. What could be the cause of this? Perhaps it would be possible to obtain significantly correlated results between antibiotic consumption in the given region and the increasing antibiotic resistance.

Unfortunately, the information on antibiotic consumption in the given region is not accessible and available. We can speculate that increased resistance of A. baumannii to antibiotics could be associated with the consumption of broad-spectrum antibiotics for the treatment of hospital infections in severely ill patients. This leads to the development of resistant strains that can persist in the hospital environment for a long time. Therefore, infection management and control are very important. Moreover, the evaluation of increased resistance based only on antibiotic consumption could be subjective as other variables such as risk factors and virulence factors could contribute to it.

Remark 8. In my opinion, the conclusions extend beyond the actual findings. I advise the authors a reconsideration of the content.

We reconsidered the conclusions as much as possible taking into the account the remarks given by other reviewers as well.

Reviewer 4 Report

In the article entitled “Associations between beta-lactamase types of Acinetobacter baumannii and antimicrobial resistance”, the authors have determined the production of different types of β-lactamases (Amp C, ESBL, KPC) in A. baumannii strains, to evaluate its association with antimicrobial resistance, and to identify the changes in these characteristics after 5 years.

1.      The conclusion in the abstract does not clearly indicate the research outcome and how this work is beneficial to the existing pool of knowledge.

2.      The formatting of the manuscript needs to be done again. The fonts are not same everywhere.

3.      Adding of graphs to increase the understandability of the data presented should be done.

4.      References need to be rechecked.

5.      A workflow diagram could be added which can enhance the readability.

Moderate editing of English language required

Author Response

Responses to Reviewer 5 Comments

 First at all, we would like to thank the Reviewer for his/her time and efforts put into reviewing our manuscript. All the remarks given by the reviewer and all the corrections done in the manuscript are highlighted in a red color. We also would like to point out that four different reviewers reviewed our manuscript and therefore we tried to make optimal corrections taking the remarks by all reviewers into consideration.

In the article entitled “Associations between beta-lactamase types of Acinetobacter baumannii and antimicrobial resistance”, the authors have determined the production of different types of β-lactamases (Amp C, ESBL, KPC) in A. baumannii strains, to evaluate its association with antimicrobial resistance, and to identify the changes in these characteristics after 5 years.

 Remark 1: The conclusion in the abstract does not clearly indicate the research outcome and how this work is beneficial to the existing pool of knowledge.

We reconsidered the conclusions as much as possible taking into the account the remarks given by other reviewers as well. Especially it is quite complicated when the opinions of the reviewers differ and when the reviewers ask to change them in contrary directions.

Remark 2: The formatting of the manuscript needs to be done again. The fonts are not same everywhere.

We apologize for these inconsistencies regarding formatting. It seems to be caused by the conversion of a Word file to the pdf one. The revised manuscript is consistent regarding the font and size.

Remark 3: Adding of graphs to increase the understandability of the data presented should be done.

Unfortunately, this could not be done as one of the reviewers asked to merge some tables and we followed this suggestion. Additional graphs would overload the manuscript, and duplication of the data by using different visual means usually is not advocated.

 Remark 4: References need to be rechecked.

The references have been rechecked.

Remark 5: A workflow diagram could be added which can enhance the readability.

We the added the following workflow diagram (please see below).

Round 2

Reviewer 2 Report

The authors tried to improve the quality of the mauscript which is encouraging; however, they failed just in the form of the manuscript which needs to be standardized (confrming the lack of rigor). In addition, most of their responses tended to explain ideas that have no relation with my comments. They also, tended to justify and insist regarding the errors observed in the manuscript.

In addition to all these comments, the manuscript needs an extensive language editing (I'm not an author to show you all errors of the manuscript, but I can show you some of them; "but also is (line11),  this study aimed to determine (line 12), comparison of the tow periods, (line 24), the Student’s t test, was used for continuous data. (173)........ see also line 177-209 (how many "A Baumanii" was repeated, line 222-223.....(these are just some example).

Briefly, the manuscript requires an extensive language editing

I have also other remarks regarding the form:

Add a reference for line 73, 74, 77

Line 169?

Line133-137: reformulate please

Reduce the size of figure 2

Delete the line from the font of figure 3 and use the same

Table: P should be p (lower case)

Try to make the valors of p in one line (exp < 0.001)

Table 4: try to make the first title in one line (Producing only one type of β-lactamase AmpC or KPC or ESBL) and correct ad AB producing...

The conclusion still be very simplistic and did no provide intresting information related to your work.

At last, I understand your hope to highlight your work by studying the evoulution of ATB in the tow periods. However, we cannot talk about evolution in the absence of data of three years (2018-2020). Thus, the use of the terms increase, decrease...is not adapted (you should just compare between the two periods). in addition you did not explain why did you not used data of this period.

In conclusion, the manuscript needs again an extensive revision. The authors should take more time to revise it and should insist to understand the comments and improve the quality of the manuscript rather than try to convince with their ideas.

 Moderate editing of English language required

Author Response

Response to Reviewer 4 Comments

Again, we would like to thank the Reviewer for his/her time and efforts put into reviewing our manuscript. All the remarks given by the reviewer and all the corrections done in the manuscript are highlighted in a red color.

Remark 1: In addition to all these comments, the manuscript needs an extensive language editing (I'm not an author to show you all errors of the manuscript, but I can show you some of them; "but also is (line11), this study aimed to determine (line 12), comparison of the tow periods, (line 24), the Student’s t test, was used for continuous data.

The whole manuscript was checked.

(173)........ see also line 177-209 (how many "A Baumanii" was repeated, line 222-223.....(these are just some example).

Corrected.

Briefly, the manuscript requires an extensive language editing

Done.

Remark 2: I have also other remarks regarding the form:

Add a reference for line 73, 74, 77

The citations were added.

Line 169?

Corrected.

Line133-137: reformulate please

The sentence was reformulated.

Reduce the size of figure 2

The size was reduced.

Delete the line from the font of figure 3 and use the same

The amendments were done.

Table: P should be p (lower case)

Done.

Try to make the valors of p in one line (exp < 0.001)

Done.

try to make the first title in one line (Producing only one type of β-lactamase AmpC or KPC or ESBL) and correct ad AB producing...

Done.

Remark 3: The conclusion still be very simplistic and did no provide intresting information related to your work.

The conclusions at the end of the text were amended. However, the conclusions in the abstract were left as they are due to limitation of the number of words for an abstract following the instructions to authors.

Remark 4: At last, I understand your hope to highlight your work by studying the evoulution of ATB in the tow periods. However, we cannot talk about evolution in the absence of data of three years (2018-2020). Thus, the use of the terms increase, decrease...is not adapted (you should just compare between the two periods). in addition you did not explain why did you not used data of this period.

The sentences containing words “increase” or “decrease” were reformulated.

Due to maternity leave of the PhD candidate (the main investigator of this study), a break in professional activities from 2018 to 2020 was taken and there was no possibility to collect and analyze A. baumannii strains during this period. Therefore, only after returning from maternity leave in 2021, the work done during 2016-2017 was continued and consequently the data were collected for years 2021 and 2022. These two periods were compared in this study.

Reviewer 3 Report

Dear Authors,

I am satisfied with the corrections which was made. In current form, the manuscript is well prepared for the publication.

Little note for the authors, the rich source of the antibiotic consumption can be found in the Antimicrobial consumption dashboard (ESAC-Net) site, highly recommended for your future studies.

Author Response

Little note for the authors, the rich source of the antibiotic consumption can be found in the Antimicrobial consumption dashboard (ESAC-Net) site, highly recommended for your future studies.

We thank the Reviewer for his/her time and effort in reviewing our manuscript, the conclusions and recommendations presented, which we will definitely consider in our future research.